# Bayesian inference across multiple models suggests a strong increase in lethality of COVID-19 in late 2020 in the UK

**Patrick Pietzonka** [1]*, **Erik Brorson**[2], **William Bankes**[3], **Michael E. Cates**[1], **Robert L. Jack**[1,4], **Ronojoy Adhikari**[1]

**1** DAMTP, Centre for Mathematical Sciences, University of Cambridge, Cambridge, United Kingdom, **2** Quantitative Research, JPMorgan Chase & Co., London, United Kingdom, **3** Applied Machine Learning and Artificial Intelligence, JPMorgan Chase & Co., London, United Kingdom, **4** Department of Chemistry, University of Cambridge, Cambridge, United Kingdom

* pietzonka@pks.mpg.de

**Data Availability Statement:** Our analysis is based on publicly available data from the UK Government (Ref. [12], metrics "newCasesBySpecimenDateAgeDemographics" and

## Abstract

We apply Bayesian inference methods to a suite of distinct compartmental models of generalised SEIR type, in which diagnosis and quarantine are included via extra compartments. We investigate the evidence for a change in lethality of COVID-19 in late autumn 2020 in the UK, using age-structured, weekly national aggregate data for cases and mortalities. Models that allow a (step-like or graded) change in infection fatality rate (IFR) have consistently higher model evidence than those without. Moreover, they all infer a close to two-fold increase in IFR. This value lies well above most previously available estimates. However, the same models consistently infer that, most probably, the increase in IFR *preceded* the time window during which variant B.1.1.7 (alpha) became the dominant strain in the UK. Therefore, according to our models, the caseload and mortality data do not offer unequivocal evidence for higher lethality *of a new variant*. We compare these results for the UK with similar models for Germany and France, which also show increases in inferred IFR during the same period, despite the even later arrival of new variants in those countries. We argue that while the new variant(s) may be one contributing cause of a large increase in IFR in the UK in autumn 2020, other factors, such as seasonality, or pressure on health services, are likely to also have contributed.

## 1 Introduction

The alpha variant B.1.1.7 of the SARS-CoV-2 virus first emerged in the UK in September 2020. It is now well known to be more infectious than the prior UK strain and for this reason, in the following months, it not only became dominant in the UK itself, but rapidly took hold in a number of other countries (including the USA) where it soon became the dominant variant [1]. However, it is less certain whether B.1.1.7 also led to more severe cases of COVID-19, resulting in turn in a higher fatality rate [2, 3]. This question has prompted the search for anomalies in the relation between aggregated case and mortality numbers. Simple visual

"newPCRTestsByPublishDate"), the Office for National Statistics [13], Our World in Data [14, 15], the French Government [16, 17], the European Centre for Disease Prevention and Control [18], and the Robert Koch Institut (Germany) [19, 20]. All relevant data can be downloaded directly at the URL's provided in these references. The study findings can be replicated in their entirety using the methods described in the article and the parameters and priors detailed in the supplementary information. The authors did not have any special access privileges to data. The data we have produced as result of the inference procedure are available as a Supporting Information file (parameters.xlsx) and in an example notebook available at https://github.com/ppietzonka/pyross-ifr-change.

**Funding:** This work was funded in part by the European Research Council under the Horizon 2020 Programme, ERC grant 740269, by the Royal Society grant RP17002, and by Microsoft Corporation through a Microsoft Research Award for the project "Building an open platform for pandemic modelling". The funders had no role in study design, data collection and analysis, decision to publish, or preparation of the manuscript. J.P. Morgan Chase \& Co.~provided support in the form of salaries for authors E.B.~and W.P., but did not have any additional role in the study design, data collection and analysis, decision to publish, or preparation of the manuscript. The specific roles of these authors are articulated in the 'author contributions' section.

**Competing interests:** The authors have declared that no competing interests exist. We received funding from Microsoft Corporation, a commercial source. E.B. is employed by a commercial company, JPMorgan Chase & Co. W.B. works for JPMorgan Chase & Co, and is employed by Kubrick Group, London. None of this does not alter our adherence to PLOS ONE policies on sharing data and materials.

inspections of these data, accounting for the typical time lag between cases and mortality, have suggested that infections with SARS-CoV-2 became more lethal in late 2020 [4].

However, the relation between reported cases and mortalities is not always straightforward to interpret. The time between infection and potential deaths is stochastic, so that a time series of mortality data will tend to show less rapid changes than corresponding case numbers. Moreover, as the available testing capacity changes with time alongside the demand for tests, which leads to a time-dependent ascertainment rate. Hence, reported case numbers are not directly representative for the true number of cases.

In order to overcome these challenges while still focusing on (nationally) aggregated data for cases and mortalities, we choose to analyse such data in the context of well-mixed compartmented models, whose complexity is adjustable for this purpose [5, 6]. These models take into account not only the stochastic dynamics of infection and progression of the disease, but also the process though which individuals get diagnosed. The latter is informed by the reported overall rate of tests performed, implicitly leading to an improving ascertainment rate if the supply of tests rises faster than the number of cases.

Our goal is to establish the evidence for a change in the UK lethality of COVID-19 in a Bayesian fashion, using the reported age-structured data for nationwide cases and mortality. For that purpose, we compare different models that either do or do not allow for a change in the infection fatality rate (IFR). We analyse differences in their posterior probability, optimised for fixed data over a set of model parameters. Generally, one would expect that the likelihood of a model increases as more details are added. In order to judge the significance of changes in the posterior, we compare a whole set of model variants differing in the level of detail, and see whether the ones that do allow for changing IFR perform better than those that do not.

Our comparison also includes model variants that would explain perceived changes in the case fatality rate merely through changes in the ascertainment rate. However, these models require implausible assumptions about changes in the testing strategy.

The model specification, simulation, likelihood computation, and optimisation is carried out using the software package PyRoss, which we have developed during the past year [5].

## 2 Model structure and data

### 2.1 Infection dynamics

We consider a suite of compartmented models (referred to as model variants below) all with $M = 7$ age groups, and 14 classes, as shown in Fig 1. These classes are abbreviated as S, E, A, Ia, Is1, Is2, Im, R; along with diagnosed/quarantined versions of all compartments (except S and E), labeled as AQ, IaQ, etc. The age groups are 0–14, 15–29, 30–44, 45–64, 65–74, 75–84, and 85+ (for our UK model variants), referred to by running indices $i$ or $j$. The transitions from the susceptible (S) to the exposed (E) compartment are induced by infected individuals in the compartments activated (A), asymptomatic (Ia), and the two symptomatic infected stages (Is1, Is2). The corresponding infection rate reads

$$\sum_j \beta_i a_i(t) C_{ij} a_j(t) [A_j + I_j^a + I_j^{s1} + c I_j^{s2}]/N_j, \tag{1}$$

with the infectiousness $\beta_i$, the intervention function $a_i(t)$, the contact matrix $C_{ij}$, the total population per age group $N_i$, and a factor $c$ for the reduction of the infectiousness in the second symptomatic stage. The contact matrix we use is based on survey data and previous Bayesian inference [7]. Quarantined individuals are assumed not to infect anyone, so the classes in Eq (1) are restricted to unquarantined compartments.

**Fig 1. Network structure of the model(s).** Every compartment (except S and E) has a quarantined version, transitions into which occur via testing. New infections are transmitted by individuals of the classes coloured red (with the stage Is2 being less infectious). Classes R and Im (mortalities) are no longer infectious.

After a presymptomatic stage (A), individuals become either symptomatically (Is1/Is2) or asymptomatically (Ia) infected, according to an age-dependent fraction $\alpha_i$ of asymptomatic cases. The outcome of the infection is either recovery (R) or death (Im). Progression though all these stages is modelled though linear transition rates, matching the latent and incubation periods, and the typical time from infection until death. The latter is determined by the exit rate $\gamma_s$ from both stages Is1 and Is2, and is inferred. Otherwise, we fix parameters relating to disease progression to values informed by the literature, in line with established literature (see Ref. [8]) and another modeling study for the UK [9]. Such fixed parameters include the rates for leaving stage E, $\gamma_E$, stage A, $\gamma_A$, and stage Ia, $\gamma_a$. The complete constitutive equations of the model are given in part A in S1 Appendix. See spreadsheet in S1 Table. for details on parameter values.

## 2.2 Contact behaviour

We model changes in the contact behaviour and (potentially) in the infectiousness via the time dependence of $a_i(t)$. These changes can be can be age-dependent, e.g. to model shielding of the elderly or increased contacts at schools and workplaces. To keep the number of control parameters manageable, we separate the time and the age-dependence of $a_i(t)$ as $a_i(t) = a(t)(1 - s(t) h_i)$ and constrain the vector $h_i$ for the age dependence to the form $[0, 0, 0, h_4, h_{5,6}, h_{5,6}, h_7]$, with the largest element set to 1 as a reference. The parameter $a(t)$ then describes overall changes in the contact frequency, while $s(t)$ describes changes in the age-structure of the contact frequency. At pre-defined (and occasionally inferred times) the parameters $a(t)$ and $s(t)$ undergo changes and their new values are inferred. We consider the interventions (or other changes) in the UK listed in Table 1. See Part B in S1 Appendix for the interventions considered for France and Germany.

## 2.3 Testing

Testing is modelled as the transfer of individuals from the undiagnosed version to the diagnosed (or "quarantined") version of a compartment. For a given overall rate of testing $\tau(t)$, the transition rate from some compartment X to XQ is

$$\tau_X = \tau(t)\pi_X\phi_X/\mathcal{N} \tag{2}$$

with normalisation $\mathcal{N} = \sum_X \pi_X X$. Here, X stands for any of the compartments S, E, A, Ia, Is1, Is2, Im, R, and the sum runs over these. Assuming perfect tests, which, however, do not detect the exposed stage, we use the false positive rate $\phi_X = 0$ for X being S, E, or R, otherwise the true positive rate $\phi_X = 1$. The factors $\pi_X$ encode testing priorities for the various compartments. We set $\pi_{Is1} = \pi_{Is2} = 1$ as a reference, and $\pi_{Im} = 20$ to ensure that mortalities get reliably detected (For the early weeks, when tests were scarce, we set $\pi_{Is2} = 5$ and $\pi_{Im} = 100$.). The only fit

**Table 1. Interventions considered in the basic UK model.**

| dates | type | control parameters |
|---|---|---|
| before 2020–03–20 | before lockdown (reference) | $a(t) = 1$, $s(t) = 0$ |
| 2020–03–20 to 2020–03–27 | imposition of lockdown | linear decrease of $a(t)$, new value $s(t)$ |
| 2020–03–27 to 2020–07–24 | easing of / increasing non-compliance with lockdown | linear increase of $a(t)$, linear change of $s(t)$ |
| 2020–07–24 to 2020–11–06 | lockdown lifted | new values of $a(t)$ and $s(t)$ |
| inferred | increase of contacts / infectiousness in autumn | tanh-shaped increase of $a(t)$, centre and width to be inferred |
| inferred | several local interventions, summarised as a single one at time to be inferred | new values for $a(t)$ and $s(t)$ |
| 2020–11–06 to 2020–12–04 | national lockdown (England) | new values for $a(t)$ and $s(t)$ |
| 2020–12–04 to 2021–01–08 | tiered lockdown | new values for $a(t)$ and $s(t)$ |
| after 2021–01–08 | national lockdown | new values for $a(t)$ and $s(t)$ |

Dates are always rounded to the closest Friday.

parameter we infer is $\pi_a$, the priority for testing individuals that are not symptomatically infected (i.e. of classes S, E, A, Ia, R, setting $\pi_S = \pi_E = \pi_A = \pi_{Ia} = \pi_R \equiv \pi_a$). It interpolates between random testing for $\pi_a = 1$ and very targeted testing for $\pi_a = 0$. The progression through stages in the quarantined compartments is the same as in the non-quarantined ones, but quarantined individuals cause no further infections (effectively assuming perfect self-isolation). Once tested positive, individuals remain in the quarantined compartments. The recovered class RQ therefore includes individuals that have actually left quarantine, but we keep this class separate for the purpose of counting previously diagnosed cases.

In summary, our model implicitly accounts for changes in the ascertainment rate, in a way that is informed by the supply (the total number of tests) and the demand (the yet untested symptomatic cases).

## 2.4 Lethality

The lethality of COVID-19 is encoded in the infection fatality rate (IFR), i.e., the probability of any infected to die eventually. As an auxiliary quantity for the specification of our model, we define a *symptomatic* infection fatality rate (SIFR), as the fatality rate for symptomatic infections only. Its numerical value can be expected to be close to the reported case fatality rate, provided that testing is exhaustively targeted at symptomatic infections.

We choose both transition rates from Is1 to Is2 and from Is2 to Im as $\gamma_s \sqrt{\text{sifr}_i}$, with the age-dependent symptomatic infection fatality rate $\text{sifr}_i$; recoveries (to class R) happen from both stages at rate $\gamma_s(1 - \sqrt{\text{sifr}_i})$. Since we use a fixed fraction $\alpha_i$ of asymptomatic cases, the IFR follows readily as $\text{ifr}_i = (1 - \alpha_i)\text{sifr}_i$. The discussion of relative changes in the IFR applies equally to the SIFR.

## 2.5 Noise

All transitions are modeled as Markov rates, i.e. they are inherently stochastic. In order to account for additional sources of noise or variation that are not present in the well-mixed

model, we infer overdispersion parameters that scale up the fluctuations in transitions related to infections, testing, and deaths.

### 2.6 Observables

We consider the reported cumulative case numbers as the sum of all quarantined classes for each age cohort. The reported cumulative mortalities are identified with the numbers of class ImQ (where the age groups 15–29 and 30–44 to match the available data). We ensure that no deaths remain unnoticed, by formally assuming that individuals in class Im continue to get tested at high priority. The numbers in all other compartments are considered as hidden, and are implicitly reconstructed by the inference procedure.

### 2.7 Data

We use weekly data from the week beginning 2020–03-07 to the week ending 2021–01-15. For cases, we use the daily numbers reported on the UK government webpage [10], reported by specimen date and in 5-year cohorts. We coarse-grain these data to weekly numbers and to our choice of age groups.

Death numbers by week of reporting have been obtained from the UK Office for National Statistics (ONS) webpage [11]. These data appear to be incomplete after 2020–12-25 and have therefore been padded with mortality data from the government webpage, which are more up-to-date but not age-structured. For the weeks in question, the sum over all age groups in class ImQ is considered as observable. (More recent data have become available in the meantime, however, we stick to this procedure to ensure comparability between results for the model variants.)

The daily number of PCR tests performed is available from the government webpage [10]. For early testing data (before 2020–04-21) we use data published by [12, 13]. The data is coarse-grained to weekly numbers, from which we build the testing rate $\tau(t)$ as a stepwise constant function.

For France, we use data for deaths in hospitals [14], and PCR testing [15]. The latter includes the number of tests performed for each age group, except for early data where instead we use non-age structured test numbers [16].

For Germany, we use data provided by the Robert Koch-Institute for cases and deaths [17], and tests [18].

## 3 Model variants

We consider several variants of the basic model outlined above, labeled by A0, A1, B0, etc. Some of the variants have an IFR that is constant in time, as indicated by the Type number 0. Type number 1 indicate time-dependent changes of the IFR; Type 2 also indicates this but via a mechanism involving slowed recovery rather than higher death rate (which also ultimately results in more deaths). The Type letters refer to other details of the model variant, *a priori* unrelated to the IFR.

### 3.1 Basic model (A0)

The model outlined above, without any further additions.

### 3.2 Step-change in IFR (A1)

As model A0, but with a simple step-change in IFR. The size of the change and time of change are inferred (except for the two youngest cohorts, where fatal cases are extremely rare). The

change in IFR is parameterised in terms of the log-ratio of the values, with a prior that is normal distributed with mean zero and standard deviation log(3). The prior for the time of the change is normal with mean 2020–12-12 and standard deviation 2 weeks.

### 3.3 Change in recovery rate (A2)

As model A0, but with a step-change of the recovery rate to $r\gamma_s(1 - \sqrt{\text{ifr}})$ with some factor $r$. The rates for progression to Is2 or Im are unchanged. This effectively changes the SIFR (and accordingly the IFR) to

$$\text{sifr}' = \frac{\text{sifr}}{\left(\sqrt{\text{sifr}} + r(1 - \sqrt{\text{sifr}})\right)^2}. \tag{3}$$

This way of introducing a change in IFR causes the death numbers to evolve more smoothly, without the need for a further fit parameter. For $r < 1$, the IFR rises, and all additional deaths occur *after* the ones that would have occurred had rates remained unchanged. (With a fixed recovery rate, the longer a patient does not recover, the more likely they are to die eventually.)

### 3.4 Model with easing/increase in infectiousness (B0)

As model A0, but with a linear increase in $a(t)$ and a linear change in $s(t)$ during the November lockdown. This could model the increasing non-compliance or changes in the infectiousness (new variant of the virus), or both.

### 3.5 Tanh-shaped IFR change (B1)

As model B0, but with a tanh-shaped change in IFR. The time around which this change is centred (which we refer to as the onset time), the width, and the amplitude are inferred. (We deem that this increased level of detail is harmonious with the already more detailed model B0.)

### 3.6 Model with change in overdispersion (C0)

As model A0, but with a change in the three overdispersion parameters for infections, testing, and deaths. The change is allowed to happen on 2020–10-02, a date chosen to match a potentially new stochastic dynamics as the second wave gains momentum. The new values of the parameters are inferred independently. This reflects potential changes in the testing strategy and in the infection dynamics in the second wave, and can avoid the overestimation of case and death numbers, that is often observed as a side-effect of mismatching overdispersion parameters.

### 3.7 Jump-like change of IFR and overdispersion (C1)

As model C0, but with a simple jump-like change in IFR.

### 3.8 Combined model (BC0)

A combination of models B0 and C0: It has a change in overdispersion parameters in the second wave and easing/non-compliance (or increasing infectiousness) during the November lockdown.

### 3.9 Combined model with changing IFR (BC1)

A combination of models B1 and C1: As model BC0, but with a tanh-shaped change of IFR as in B1.

### 3.10 Test and trace (TT0)

When case numbers are low, effective contact tracing is possible. This could mean that more asymptomatic cases are uncovered in summer than at the height of the first and second wave. As a simple model for this effect, building on model BC0, we allow for the inference of the testing priorities $\pi_A$ and $\pi_{Ia}$ for pre- and asymptomatic infected individuals different from the priority $\pi_a$ of classes S and R. This change comes into effect with the beginning of large-scale contact tracing on 28th May 2020. Testing priorities remain unchanged thereafter; however, as long as the testing priorities of A and Ia remain below those of Is, the effect of contact tracing will only become relevant for large test rates and low case numbers, so that the class Is of undiagnosed individuals can be depleted.

### 3.11 Test and trace with changing IFR (TT1)

As model TT0, but with a tanh-shaped change in IFR (as in B1).

### 3.12 Time-dependent test priority (P0)

As model BC0, but with a time-dependent change in $\pi_a$, the only parameter entering our model for testing. The change in $\pi_a$ is tanh-shaped, with centre (the onset time), width and amplitude to be inferred. This change may reflect changes in the testing strategy, that have happened during the course of the pandemic.

## 4 Model comparison

Using our software package PyRoss, we can calculate the logarithmic likelihood of the observed data for each of the model variants and for any choice of the model parameters and initial conditions [6]. This computation is based on the inherent stochasticity specified for the model. It employs a multivariate Gaussian approximation of the joint probability of all compartment values at all observed points in time, thus taking into account all correlations across compartments and time. The approximation becomes exact in the limit of large population sizes (provided that the data is sufficiently close to the mean) and is therefore appropriate for our well-mixed models applied at national level.

Given an informed choice of prior distributions for all parameters and initial conditions, we have determined for each model variant the parameters that maximise the posterior probability (MAP). To decide between different modelling hypotheses, a rigorous approach is to compute the model evidence (also known as the marginal likelihood). In this Section we take the simpler approach of comparing the log-posterior probabilities—to a first approximation, models with large posterior probability should be preferred. However, that approach can suffer from over-fitting: that issue can be addressed via estimates of the evidence. This point is addressed in Sec 5, below. (Anticipating the answer, we find that the conclusions of this Section—based on the log-posterior—are robust.)

Results are summarised in Table 2 and Fig 2. Remarkably, the model variants with a change in IFR consistently attain higher (non-normalised) posteriors. As we have centred the prior for the factor of change in IFR to the value 1 (representing the null-hypothesis that there has been no change), the prior always decreases with a change in IFR, but this reduction of the prior is overbalanced by quite some margin by the increased likelihood derived from data. The

**Table 2. Summary of MAP results.**

| Country | Model | # Params | log-Prior | | log-Posterior | | log-Evidence | | IFR change | |
|---|---|---|---|---|---|---|---|---|---|---|
| | | | abs | Δ | abs | Δ | abs | Δ | Onset | Factor |
| UK | A0 | 67 | −354 | | −4281 | | −4149 | | | |
| | A1 | 69 | | −20 | | +55 | | +50 | 27 Oct (±2 d) | 1.97(±0.11) |
| | A2 | 69 | | −9 | | +49 | | +44 | 24 Nov (±8 d) | $r = 0.40(±0.07)$ |
| UK | B0 | 69 | −350 | | −4269 | | −4147 | | | |
| | B1 | 72 | | −21 | | +61 | | +64 | 8 Nov (±7 d) | 2.14(±0.17) |
| UK | C0 | 70 | −365 | | −4282 | | −4150 | | | |
| | C1 | 72 | | −35 | | +52 | | +63 | 28 Oct (±2 d) | 2.02(±0.12) |
| UK | BC0 | 72 | −400 | | −4272 | | −4139 | | | |
| | BC1 | 75 | | −26 | | +56 | | +56 | 9 Nov (±7 d) | 2.20(±0.18) |
| | P0 | 75 | | −22 | | +59 | | +69 | 29 Oct (±5 d)[a] | [3.000][a,b] |
| UK | TT0 | 73 | −368 | | −4270 | | −4153 | | | |
| | TT1 | 76 | | −25 | | +56 | | +57 | 9 Nov (±7 d) | 2.20(±0.19) |
| GER | C0 | 59 | −268 | | −3438 | | −3376 | | | |
| | C1 | 61 | | +3 | | +79 | | +77 | 26 Nov (±2 d) | 1.80(±0.08) |
| FRA | C0 | 75 | −254 | | −4137 | | −4121 | | | |
| | C1 | 77 | | 0 | | +55 | | +58 | 3 Nov (±1 d) | 1.37(±0.11) |

We list the country considered along with the model variant, the number of inferred parameters and initial conditions, the logarithmic prior, posterior, and model evidence, and, if applicable, the inferred onset time and factor of a change in IFR. For Type 0 models without any IFR change (printed in bold), absolute values of log-prior, log-posterior (non-normalised), and the logarithmic model evidence are shown, indicated by "abs". For other models, we show values relative to the corresponding base model, indicated by Δ. The indicated uncertainties in the inferred parameters for the IFR change correspond to a single standard deviation in a Gaussian approximation of the posterior.

[a] Changes in $\pi_a$

[b] MAP value has attained an upper bound set by prior

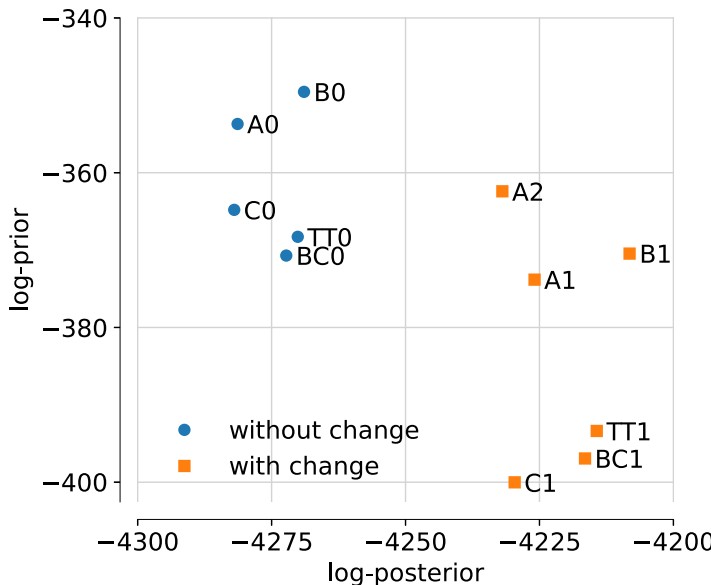

**Fig 2. Plot of log-posterior and log-prior.** Models without change in IFR in blue, models with change in IFR in orange.

inferred change in IFR is always around the factor of two for the UK data. Note that for model A2 the change in the recovery rate amounts even to a factor of 2.35 in IFR for the oldest cohort.

It is remarkable that even though the models A0, B0, C0, and BC0 have different numbers of fit parameters, their variation in the log-posterior is nowhere near as big as the difference to the variants with a changing IFR. This observation already suggests that the increase in likelihood for the variants with changing IFR is not a mere consequence of overfitting due to the additional parameters.

For each model and set of MAP parameters, we can plot a deterministic solution. This is the most likely trajectory, conditional on the inferred initial condition, and the mean of the multivariate Gaussian approximation for all compartment values at all times. Fully detailed plots of the MAP trajectories for each of the model variants are shown in part C in S1 Appendix.

Unsurprisingly, the results for the most detailed model variants BC0 and BC1 produce mean trajectories for cases and mortalities that fit the data best, as shown in Fig 3. Among these two solutions, the even better fit of the model variant with change in IFR (BC1) is not only evident through the likelihood computation, but also visible to the naked eye.

In Fig 4 we show the MAP trajectories for the variants of models A, B and C. On visual inspection, none of these models fit the data as accurately as BC0 and BC1 in Fig 3. However, such visual inspection is not always as reliable an estimate as our Bayesian posterior, which accounts for temporal correlations in the data. For example, we find that the likelihood strongly penalises models where the rate of growth (or decay) of infections does not match the data. Also, the (idealised) step-changes in the intervention function $a(t)$ mean that MAP trajectories may over/under-shoot the data at the change points. In combination, these two factors mean that agreement between expected trajectory and data may be imperfect on visual inspection; however, the posterior is being correctly maximised and the results of inference are robust. This is because deviations of the data from the mean trajectory are consistently taken into account for the implicit inference of unobserved compartments. For example, the inferred change in IFR is remarkably consistent between the model variants for the UK, despite considerable differences in the mean trajectory for each model. This observation reassures us that the evidence for a change in IFR would persist in a model that is even more detailed than model BC.

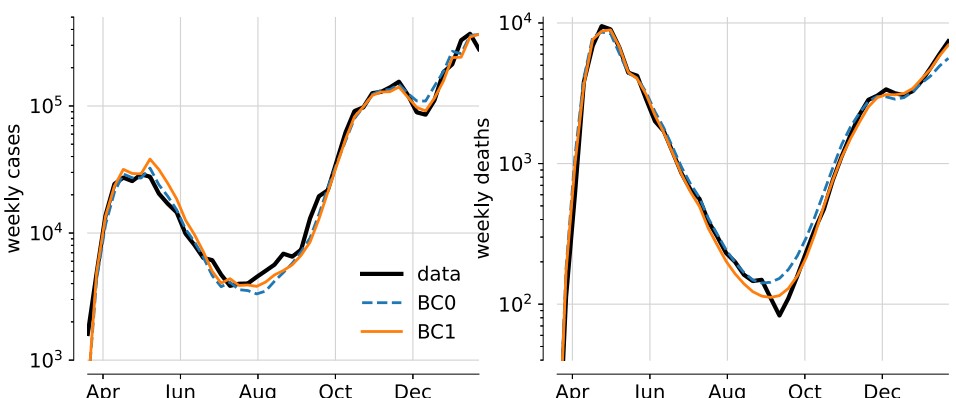

**Fig 3. Mean trajectories of weekly cases and deaths for model variants BC0 and BC1.** Deterministic trajectories for the MAP parameters of models BC0 (dashed blue) and BC1 (solid orange), along with data (black).

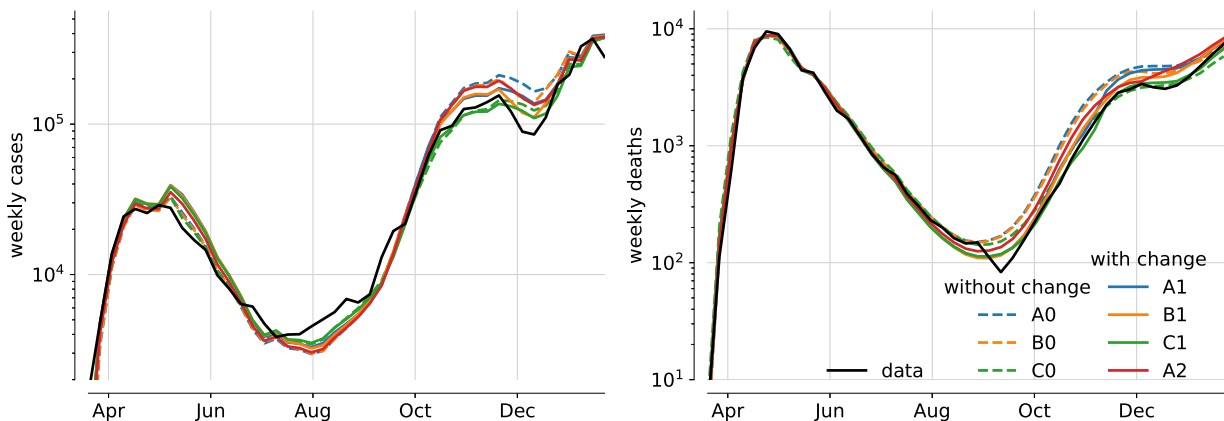

**Fig 4. Mean trajectories of weekly cases and deaths for further model variants.** Deterministic trajectories for the MAP parameters of the various model variants, along with data (black). Models without change in IFR are shown as dashed, models with change in IFR solid.

Some differences between models A/B and C are due to a generic feature of our computation of the likelihood. When the estimated overdispersion parameters are too low to account for the observed noise in the data, the optimiser tends to overestimate the expected case and deaths numbers, thereby increasing the variance of weekly changes. Considering the resulting MAP trajectories for models A and B, it seems that the inferred overdispersion parameters, which serve well to fit the first wave, are too small to match the level of noise in the second wave. This leads to an overestimation of both expected case and death numbers. Differences between cases and deaths in this overestimation could negate the perceived change in IFR. This has prompted us to analyse model C, allowing for changes in the overdispersion parameters for the second wave. It is consistent with the results of models A and B, which rules out that the observed changes in the IFR stem from temporal changes in the overdispersion.

The models C0/C1 reproduce the observed height of the second wave, but not the dip between the second and third wave. Note that model C (just like model A) has the November lockdown fixed without easing, leading to a larger reduction in cases and deaths than in reality. Model BC (easing and change of overdispersion) reproduces the short-livedness of this reduction better.

The sudden drop in mortalities (individuals with COVID-19 mentioned on the death certificate) in late August / early September is not reproduced by any of our models. This might be related to changes in the legal definition of such deaths. We do not model this here, but note that adjusting the data for a lasting change in the definition from this time onwards would likely lead to an even larger increase in the IFR than the MAP estimates 1.9–2.2 reported above.

We note that for the exit rate from the two stages of symptomatic infection, the value $\gamma_{Is} \approx$ 0.43 (per week) is inferred for model BC1 (and similar values for the other UK model variants). This corresponds to a mean time from onset of symptoms to death of approximately 33 days, which is considerably more than the 18 days reported in the cohort based study of Ref. [19]. This discrepancy could be due to reporting delays in the mortality data of Ref. [11]. Also, this source includes all deaths *involving* (rather than *due to*) Covid-19, which may include deaths related to later complications.

We also did the inference procedure for model C0/C1 with data for France and Germany, using appropriate forms of the intervention function, as detailed in part B in S1 Appendix. A similar change in IFR seems to be present there as well, though somewhat less pronounced in France and happening somewhat later in Germany.

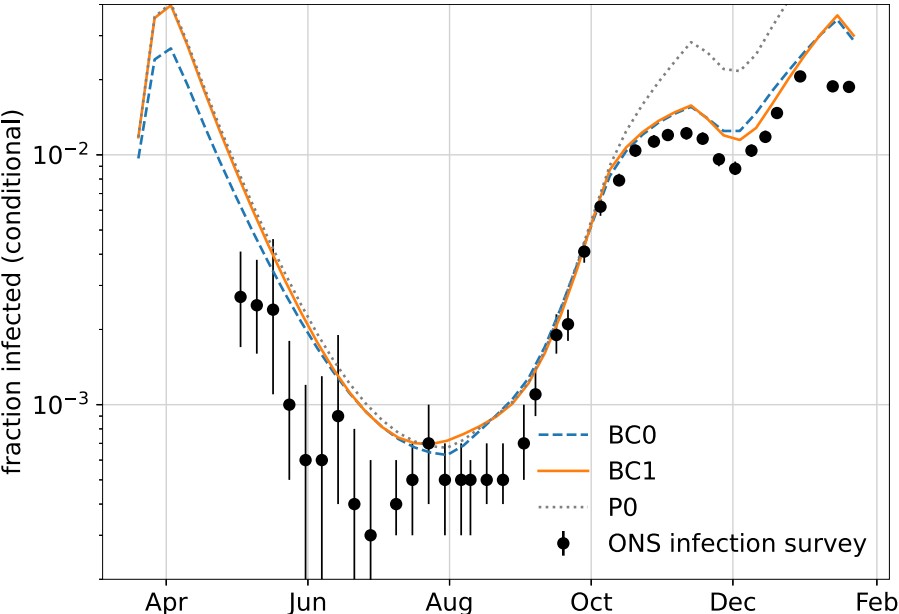

**Fig 5. Estimated fraction of infected population conditional on observation.** Mean fraction of people infected (symptomatic and asymptomatic) in the total population, conditional on the observation of cases and deaths in models BC0 (blue), BC1 (orange) and P0 (grey). For comparison, we also show the prevalence of infections reported in the ONS infection survey.

In Fig 5 we show the inferred mean fraction of infected individuals, conditional on the observation of cases and deaths at all points in time. This result is compared to the data from the ONS infection survey [20], seeking to determine the fraction of the infected population through random asymptomatic testing. We note that numbers in our results are always bigger than in the infection survey, except when they are rapidly rising. This could be an indication that individuals count as infected for longer in our models than they test positive in a PCR test. Also, the fixed estimate we use for the age-dependent fraction of asymptomatic infections may be incorrect (based on early data [21], as described in our paper [6]).

We note that differences in the conditional numbers of infections between models BC0 and BC1 mainly show up in the first wave. For that time, data from the infection survey is not available, and data on testing may be incomplete. Nonetheless, it is remarkable that the inferred numbers of infection largely agree between models BC0 and BC1 from June onwards, encompassing the inferred time of change in IFR. Hence, we can rule out that the change in IFR in model BC1 is merely due to changes in the inferred true number of cases. It is rather that the observed timeline of deaths is more likely in model BC1 than in model BC0, for similar estimated total numbers of infections.

Based on the remaining discrepancies between the our inferred infection numbers and the ONS survey, one could still argue that we overestimate the true case numbers early on and/or underestimate them later, leading to an apparent increase in the IFR. The model variants TT0/TT1 and P0 serve to address this possibility.

The goal of contact tracing is to detect and isolate asymptomatic cases of COVID-19 and ideally to also detect cases early on in the presymptomatic stage. However, due to limited capacity, the test and trace system is only effective when case numbers are low and there is sufficient testing capacity, such as in the summer months. This could mean that with increasing testing numbers and the large-scale test and trace system being put in place in late May, the

reported cases after the first wave are closer to the true cases than expected by the model variants considered so far. The results for the pair of variants TT0 and TT1, with their (albeit rudimentary) realisation of contact tracing, give no indication that this might explain an apparent change in the IFR. Quite to the contrary, TT1 infers a somewhat larger change in IFR than the Type 1 models already considered. The posterior of models TT0 and TT1 change only marginally compared to BC0 and BC1, respectively.

In principle, it is possible that the inferred factor-two change in IFR in early November could instead be explained by large changes in the numbers of undiagnosed cases. We illustrate this fact using model P0. It allows for a time-dependent change of the testing priority $\pi_a$, and we deliberately set a loose prior on the timing and amplitude of this change. This model attains a posterior probability that is comparable to that of the models with a change in the IFR. (Coincidentally, prior and posterior are almost identical to those of TT1, therefore P0 is not shown in Fig 2.) The prior would have allowed to place the change in $\pi_a$ as early as July, yet it is inferred as late November (four weeks *after* the mean of the Gaussian prior distribution). The amplitude of the change in $\pi_a$ is inferred as a factor 3, unexpectedly saturating an upper bound we set on this parameter. This would mean that, to explain the data in terms of a changed testing regime rather than an actual IFR change, at the height of the second wave, tests must have become at least three times *less* targeted at symptomatic cases of COVID-19. It seems to us highly unlikely that this change could be explained, for example, by a rise in cases of flu, and we would expect evidence for this scenario to rapidly diminish under a more realistic choice of prior. We also note that, as shown in Fig 5, the inferred true number of infections for P0 matches the data from the infection survey much worse than for models BC0 and BC1.

## 5 Model evidence and parameter uncertainties

This section refines the model comparison by using an estimate of the Bayesian model evidence (also known as the marginal likelihood). The core of our analysis is the Bayes theorem

$$P(\boldsymbol{\theta}|\text{data}) = \frac{P(\text{data}|\boldsymbol{\theta})P(\boldsymbol{\theta})}{Z(\text{data})}. \tag{4}$$

Here $P(\boldsymbol{\theta}|\text{data})$ is the posterior of the parameters $\boldsymbol{\theta}$ conditional on the observation of the data, $P(\text{data}|\boldsymbol{\theta})$ is the likelihood of the data calculated in a specific model, and $P(\boldsymbol{\theta})$ is the prior. The product $P(\text{data}|\boldsymbol{\theta})P(\boldsymbol{\theta})$ is quoted as non-normalised posterior for the MAP parameters $\boldsymbol{\theta}^*$ in Table 2 and Fig 2. Its normalisation $Z(\text{data})$ is the model evidence, or marginal likelihood [22–24].

For a complete analysis of the model evidence, a sampling of the posterior over the parameter space would be necessary, e.g. using Markov Chain Monte Carlo (MCMC) techniques [6]. Since the evaluation of the likelihood for the given amount of data and complexity of the model is computationally rather expensive, we refrain from the MCMC sampling and instead work with a local Gaussian approximation of the posterior around the MAP parameters,

$$P(\boldsymbol{\theta}|\text{data}) \approx P(\boldsymbol{\theta}^*|\text{data})\, e^{-(\boldsymbol{\theta}-\boldsymbol{\theta}^*)\cdot\boldsymbol{H}(\boldsymbol{\theta}-\boldsymbol{\theta}^*)/2}. \tag{5}$$

Here, $\boldsymbol{H}$ is the Hessian matrix of the (negative) log-posterior at $\boldsymbol{\theta} = \boldsymbol{\theta}^*$, calculated using finite differences. The Gaussian approximation can be validated by comparing it to the computed posterior along "slices" of the parameter space where only one parameter is varied while others are held constant, as exemplified in Fig 6.

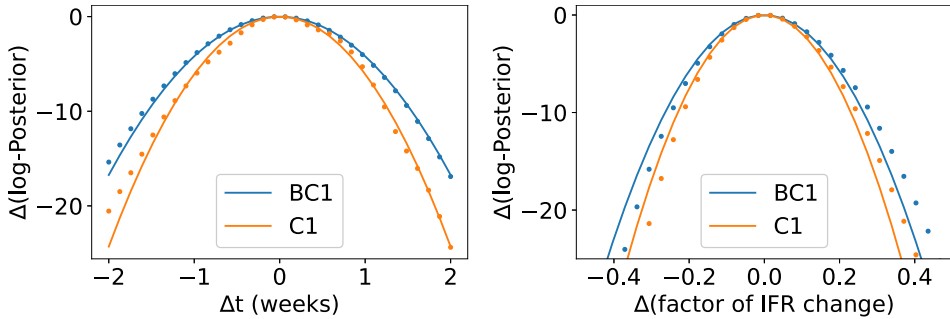

**Fig 6. Logarithm of the posterior distribution along a slice of the parameter space.** Data for model variants BC1 (blue) and C1 (orange) are shown. The computed posterior is shown as points, the Gaussian approximation of Eq (5) is shown as solid lines. We vary the onset time (left) and the factor (right) of the change in IFR around the value of $\boldsymbol{\theta}^*$, keeping all other parameters fixed. Values shown for the log-posterior are the difference to the maximal values for models BC1 and C1, respectively.

The logarithmic model evidence follows as

$$
\begin{aligned}
\log Z(\text{data}) \quad &= \log[P(\text{data}|\boldsymbol{\theta}^*)P(\boldsymbol{\theta}^*)] + \log \int d\theta \, \frac{P(\text{data}|\boldsymbol{\theta})P(\boldsymbol{\theta})}{P(\text{data}|\boldsymbol{\theta}^*)P(\boldsymbol{\theta}^*)} \\
&\approx \log[P(\text{data}|\boldsymbol{\theta}^*)P(\boldsymbol{\theta}^*)] + \frac{1}{2}\log[(2\pi)^k \det \boldsymbol{H}],
\end{aligned}
\tag{6}
$$

with $k$ being the number of inferred parameters. In the second line, we have used the Gaussian approximation (5) to evaluate the integral. The results for the model evidence for each of the variants are listed in Table 2. The comparison of model evidences supports the conclusions drawn in the previous Section. The first term in Eq (6) is the posterior at the MAP parameters, which we have so far used as a proxy for the evidence. This term tends to dominate the integral for $\log Z$, but the second term is important for penalising over-fitting. In conclusion, we see clearly that the inferred evidence for changes in the IFR are not due to over-fitting.

The local Gaussian approximation of the posterior can also be used to estimate uncertainties in the inferred parameters. The inverse $\boldsymbol{H}^{-1}$ is the covariance matrix of the inferred parameters in the posterior distribution. The square root of its diagonal elements yields a single standard deviation (or 68% credible interval) for each individual parameter, such as the ones quoted in Table 2. (Note that the computation of the inverse Hessian performs the marginalisation over all other parameters for every diagonal element.) In the supplementary information, we provide a spreadsheet listing all inferred parameters and uncertainties (see S1 Table). We note that a small standard deviation should not be mistaken for certainty about a particular parameter in general, if otherwise the evidence for the respective model variant is low. For instance, we notice that the uncertainty in the onset date is significantly smaller for models with a step-like change in the IFR (model variants A1, C1) than for the ones with a smooth change. This could be due to the likelihood of these models being more sensitive to short-lived weekly fluctuations in the data, fitting some sudden, albeit transient, increase in reported deaths.

Moreover, we can explore the posterior distribution of the IFR before and after the change by drawing samples from the multivariate Gaussian distribution of the parameters describing the IFR. The comparison between models BC0 and BC1 in Fig 7 shows that the IFR inferred for BC0 lies in between the early and the late value inferred for BC1, regardless of the age group. Moreover, we see that the difference between the values before and after the change is

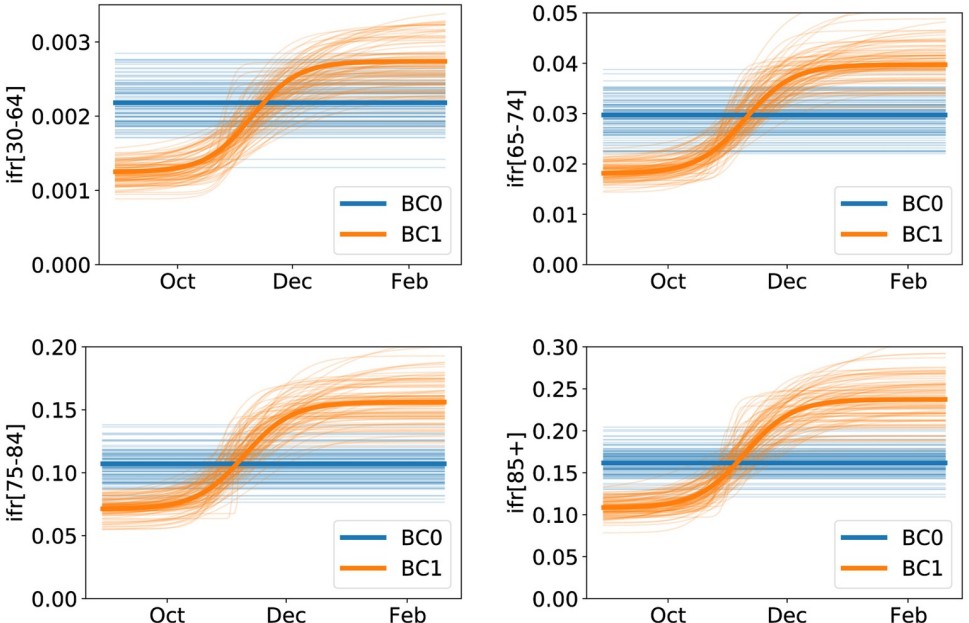

**Fig 7. Inferred changes of the IFR.** We show data for model variants BC0 (blue) and BC1 (orange, time-dependent), for four different age groups (indicated in square brackets). The thick lines represent the IFR determined from the MAP parameters, the thin lines correspond to 100 samples of the parameters for the IFR drawn from the Gaussian approximation of the posterior.

bigger than the uncertainty in any single value for the IFR. For the age-group 75–84 the IFR increases from 7(±1)% to 16(±2)% and for the age-group 85+ from 11(±1)% to 23(±2)%. The estimates for these absolute values of the IFR are in line with other reports in the literature [25–32]. Note, however, that the absolute estimates (unlike the relative change) ultimately depend on the age-dependent fraction of asymptomatic cases, which we keep fixed throughout the inference procedure.

As a final note, we consider whether the observed evidence for a change in the IFR could be affected by confirmation bias. We had been motivated by previous reports, based on a non-Bayesian analysis [2–4, 33]. We were specifically interested in changes in the IFR around the time when the alpha variant became dominant, and we were deliberately setting prior for the time of IFR change such that we would not pick up other, seasonal changes. In practice, however, the 2-week standard deviation in the prior does not impose a strong bias towards a specific date. For instance, a 4-week deviation from the prior mean would correspond to a change in the logarithmic prior of just two, this is very little compared to the changes in posterior and evidence that qualify the model variants with changing IFR. Moreover, for the models for France and Germany we had no such prior information, nonetheless we detect similar evidence for a changing IFR.

## 6 Discussion

In this paper, we have reported evidence for an increase in lethality of COVID-19 in the UK in late autumn 2020. Bayesian inference provides clear and consistent evidence for such an increase, across the suite of models we considered. This finding complements similar conclusions based on the visual inspection of nationally aggregated case and mortality data [4].

Our approach uses the same kind of data, but we use models that differentiate between reported cases and the true number of cases. They are informed by the reported number of tests performed, implicitly accounting for a time-dependent ascertainment rate. Estimates of the absolute value of the IFR are always beset with uncertainty, due to an unconstrained dark figure of less severe or asymptomatic infections [25, 30]. Nonetheless, evidence for a *relative* change in the IFR shows up clearly in our model comparison.

We would not generally expect aggregated case and mortality data, analysed with a well mixed compartmental model (or suite thereof), to identify clear or definitive causes for an increase of this kind. So our conclusion is (in common with [4, 33]) that something happened to increase lethality, but the data does not clearly say what it is that happened.

It is natural to speculate that the increase is related to the emergence of one or more new virus strains, whose potentially increased lethality has been the subject of several cohort-based studies summarised previously in a UK Government publication [3]. Those works estimate a lethality increase for B.1.1.7 by factors ranging up to 1.7 [34]. In contrast, a study by Davies *et al.* using a compartmented model including two virus strains [35] did not produce clear evidence for any change in severity (only a large increase transmissibility). From our suite of compartmented models, we infer not only a larger increase in IFR than so far attributed to B.1.1.7 severity alone, but also a most likely onset date for this increase that *precedes* the widespread emergence of B.1.1.7 in the UK. Moreover, we infer a very similar change in Germany (alongside a more modest one in France) despite even lower prevalence there, at that time, of B.1.1.7 or any comparable new strain. We argue on these grounds that the arrival of B.1.1.7 is unlikely to be the sole cause of the dramatic UK increase in IFR in late 2020.

This suggests that factors such as seasonality and/or pressure on health services may have contributed to the change. Of these, seasonality more credibly would have had similar effects in the UK and Germany (and perhaps weaker effects in France, but this is far from clear). This conclusion may also be supported by a study from Israel addressing changes of in-hospital mortality rates [36]. If such factors started to increase IFR in early autumn, then the MAP estimates of the onset time, for either a single step-change or a single ramped increase, might represent a compromise between two separate episodes of increasing IFR, with only the latter linked to a new variant. This could allow the fitted onset time to precede arrival of B.1.1.7, with the latter still contributing strongly to the total increase in fatality.

Such arguments remain, for now, speculative. To further investigate potential connections between new variants and the observed change in IFR, one should explicitly represent the dominant mutant strain though additional model compartments, with increased infectiousness and possibly increased lethality. Such a model could be calibrated using data for the prevalence of mutant strains.

The increase in lethality might also be generic for the peaks of waves of infection, when hospitals are under severe strain. We have not yet analysed the possibility for *two* changes in IFR: a reduction in spring and an increase in autumn. This might also reflect seasonal variations in the resilience of patients. Figs 3 and 4 show that the inferred most likely trajectories actually overestimate the cases in the first wave somewhat in order to get the number of deaths right, and even more so for the models with change in IFR. The possibility of a high IFR in the first wave, falling back in late spring before rising again in autumn, might resolve this discrepancy.

It will be interesting to see whether the evidence for a change in IFR persists as models become calibrated with more recent data and additional types of observations. This might include data for hospital admissions, antibody testing, or random asymptomatic testing (as already considered *a posteriori* in this report). More data would also enable more detailed versions of our models to be constructed. For example, we have neglected so far the possibility of reinfection (loss of immunity). It would be also possible to represent more accurately the

timing of the progression through stages of infection [37]. Large-scale vaccination campaigns may also be accounted for in future variants of these models.

As an additional caveat, we emphasise that these well-mixed models cannot describe regional variability, such as situations where the epidemic is shrinking in one region and growing in another. Moreover, the evidence for the UK IFR change is partly based on the relatively poor fits achieved by models with fixed IFR. It follows that if substantially improved fits were obtained for models that include regional variability (with fixed IFR), our conclusions might have to be re-evaluated. On the other hand, we are not aware of a specific mechanism by which regional variability would generate the discrepancies observed here between the models and the data. These possibilities might be also tested by future work.

Finally, we would like to advocate the advantages of creating a suite of models within a single platform to which consistently identical inference procedures can be applied. It is of course valuable to have independent modelling teams doing their own preferred type of data analysis and then comparing the results of these studies to see if a consensus emerges. However, the integrated model-building and inference machinery gathered within the PyRoss platform has in this case allowed rapid implementation of a number of purpose-built models and their comparison, in a fully Bayesian way, in the month immediately following the appearance of [4]. (In fact [4] was posted on medRxiv on 22 January; a complete draft of the current paper was circulated to relevant members of the UK Government advisory group SPI-M on 24 February.)

The PyRoss platform [38] is open source, and freely available to all users.

## Supporting information

**S1 Appendix. Model details and plots.** Part A: Constitutive equations for the basic model. Part B: Intervention functions for Germany and France. Part C: Detailed plots of the MAP trajectories.
(PDF)

**S1 Table. Spreadsheet of parameters and inference results.** Sheet 1: Overview of key parameters and results. Sheet 2 (3, 4): Detailed information on parameter prior and posterior values and uncertainties for all UK (GER, FRA) model variants.
(XLSX)

## Acknowledgments

We thank Graeme Ackland, Daan Frenkel, and Julia Gog for helpful discussions. We also thank William Peak and Andrew Ng from JPMorgan Chase & Co. and all PyRoss contributors (see [5]), in particular Julian Kappler, Yuting I. Li, Paul B. Rohrbach, Rajesh Singh, and Günther Turk.

## Author Contributions

**Conceptualization:** Robert L. Jack, Ronojoy Adhikari.

**Data curation:** Patrick Pietzonka, Erik Brorson, William Bankes.

**Formal analysis:** Patrick Pietzonka, Erik Brorson, William Bankes.

**Funding acquisition:** Michael E. Cates, Ronojoy Adhikari.

**Methodology:** Patrick Pietzonka, Erik Brorson, William Bankes, Michael E. Cates, Robert L. Jack, Ronojoy Adhikari.

**Project administration:** Michael E. Cates, Ronojoy Adhikari.

**Software:** Patrick Pietzonka, Erik Brorson, William Bankes.

**Writing – original draft:** Patrick Pietzonka.

**Writing – review & editing:** Erik Brorson, William Bankes, Michael E. Cates, Robert L. Jack, Ronojoy Adhikari.

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
