## [Decision Letter · Decision Letter 0]

26 Apr 2021

PONE-D-21-08131

Bayesian inference across multiple models suggests a strong increase in lethality of COVID-19 in late 2020 in the UK

PLOS ONE

Dear Dr. Pietzonka,

Thank you for submitting your manuscript to PLOS ONE. After careful consideration, we feel that it has merit but does not fully meet PLOS ONE’s publication criteria as it currently stands. Therefore, we invite you to submit a revised version of the manuscript that addresses the points raised during the review process.

Please respond to the reviewer comments on a point-by-point basis and revise the manuscript accordingly.  

I'd also like the authors to explore whether fixing most of the \\pi parameters, e.g. \\pi_{Is}, affects the results.  This seems to assume that testing rates among the symptomatic didn't change over time, or merely that those changes will all be handled by the \\pi_a term (which really appears intended to represent tracing and capture of asymptomatics).  Is this assumption a problem for constraining fluctuating testing capacity and demand for testing rates over time?  Evidence suggests that ascertainment rates increased over time, which would work against the rising IFR reported here.  What do the raw case fatality rates show (when convolving cases forward to deaths)?  Do they rise as well?  How about the hospitalization fatality rates?  Do they rise or fall in the study period?  This can be pulled from observations and provide an indication as to whether clinical treatment has improved.  Is your main finding the result of a poorly constrained total number of infection (i.e. the ascertainment rate)?

We look forward to receiving your revised manuscript.

Kind regards,

Jeffrey Shaman

Academic Editor

PLOS ONE

Journal Requirements:

"This work was funded in part by the European Research Council under the Horizon 2020 Programme, ERC grant 740269, by the Royal Society grant RP17002, and by Microsoft Corporation through a Microsoft Research Award for the project "Building an open platform for pandemic modelling". The funders had no role in study design, data collection and analysis, decision to publish, or preparation of the manuscript."

We note that you received funding from a commercial source: Microsoft Corporation.

We note that one or more of the authors are employed by a commercial company: JPMorgan Chase & Co.

3.1. Please provide an amended Funding Statement declaring this commercial affiliation, as well as a statement regarding the Role of Funders in your study. If the funding organization did not play a role in the study design, data collection and analysis, decision to publish, or preparation of the manuscript and only provided financial support in the form of authors' salaries and/or research materials, please review your statements relating to the author contributions, and ensure you have specifically and accurately indicated the role(s) that these authors had in your study. You can update author roles in the Author Contributions section of the online submission form.

3.2. Please also provide an updated Competing Interests Statement declaring this commercial affiliation along with any other relevant declarations relating to employment, consultancy, patents, products in development, or marketed products, etc.  

5. Please ensure that you refer to Figures 6-21 in your text as, if accepted, production will need this reference to link the reader to the figure.

6. We note you have included a table to which you do not refer in the text of your manuscript. Please ensure that you refer to Tables 3 and 4 in your text; if accepted, production will need this reference to link the reader to the Table.

Reviewers' comments:

Reviewer's Responses to Questions

**Comments to the Author**

1. Is the manuscript technically sound, and do the data support the conclusions?

Reviewer #1: Yes

Reviewer #2: Partly

2. Has the statistical analysis been performed appropriately and rigorously? 

Reviewer #1: Yes

Reviewer #2: No

3. Have the authors made all data underlying the findings in their manuscript fully available?

Reviewer #1: Yes

Reviewer #2: Yes

4. Is the manuscript presented in an intelligible fashion and written in standard English?

Reviewer #1: Yes

Reviewer #2: Yes

5. Review Comments to the Author

Reviewer #1: The paper was a pleasure to read. The summary is excellent. The clear way in which you present your models and your results gives me total confidence in your work. I recommend that it should be published. It is valuable not only for its results but also because it provides a nice demonstration of your methods and of your PyRoss software package and will help others to build on this. I only have two suggestions 1) please make the code available online and 2) it would be nice if you could give the reader access to the MAP values for all inferred parameters. I would have liked to check whether they look believable, but I also think I would have learned something from them.

Reviewer #2: Dear editor,

the manuscript analyses public COVID-19 data in the UK, France and

Germany using a SEIR type compartmental model. The text is clearly

written and the proposed models interesting and well thought-out.

The authors conclude that the IFR in the UK and Germany was higher

in the end of 2020 by a factor of around 2 when compared to the

first semester. They also conclude that this increase precedes the

widespread circulation of the new major SARS-CoV-2 variant B.1.1.7.

I consider this a very strong claim which is not convincingly

backed by the methods and analyzed data presented in the current

manuscript. I cannot recommend publication unless the authors are

able to provide further details and/or revise their analysis in

order to better substantiate their main conclusion.

I explain below both my major and minor concerns in detail.

MAJOR POINTS

1 - The authors make use of the PyRoss package in order to evaluate

their posterior and quote maximum posterior values for their

parameters. Although I understand this package was already

discussed in more detail in a previous manuscript, the authors

should explain here in more detail the methods involved in their

analysis. For instance, are the quoted values for the IFR change

marginalized over all other model parameters? Was the posterior

sampled with MCMC methods or is it the maxima found with a simpler

scheme? If the latter, what is the justification and how accurate

is it?

2 - Why do the authors refrain from quoting confidence intervals CI

(ideally highest density intervals) in most of their results? The

claim that "the calculation is tedious" seems unjustifiable, as the

discussed method based on the Hessian seems to be the very well

established method of using the Fisher information Matrix. In

particular the Li et al. 2020 paper which describes the package

PyRoss discusses the implementation of both MCMC and Fisher

information Matrix methods. In order to help make sense of the main

results for the IFR I consider estimations of the marginalized CI

for all main results imperative. Ideally one should show plots of

the posterior for the IFR (before, after and mean IFR for the

models without a change) for at least some models. Also, in the 2

cases for which uncertainties are shown, they seem quite narrow,

specially for the German IFR factor. Is that expected after

marginalization over dozens of parameters?

3 - The authors choose not to make a full model-comparison using

the Bayesian evidence and instead rely on a simple analysis of the

maxima of the Posterior. I think that if the full evidence cannot

be computed due to computational complexity, at least some better

quantitative proxy for it should be employed. Alternatives are the

Akaike or Bayesian Information Criteria, which are fast to compute.

Or at the very least a Goodness of Fit test.

4 - The main conclusion is that the IFR seems to have increased by

a factor of around 2 in the end of 2020. Nowhere in the text

however are the actual inferred values for the IFR (before and

after) written down. Nor is there a discussion on how these

estimates compare with other IFR estimates in the literature (for

the UK or other countries), specially with those which do not rely

on similar modelling.

5 - As the authors claim, for IFR estimates one needs accurate

estimates of the total number of cases, including undiagnosed ones.

As the authors point our a bias in the estimated number of cases

indifferent months can affect their inferred IFR increase. This

point deserves a more careful discussion as any IFR estimate hinges

strongly on the estimation of the total number of cases. I would

like the authors to discuss in more detail how the ONS random

asymptomatic testing was conducted and how reliable are their

estimates. How does it compare with other random seroprevalence

surveys (in the UK or elsewhere)?

MINOR POINTS

1 - There have been estimates in the literature of the time lag

between contagion and mortality and between development of symptoms

and mortality. How do these estimates compare with the values in

the models used?

2 - Some variables are not clearly defined in the text, which may

hinder understanding readers less familiar with SEIR models. For

instance: phi_X, gamma_A, gamma_E, a(t), s(t).

3 - For the models with allow a step-change in IFR values the

window for change has a narrow 2-week sigma. Is it possible that by

using a much broader window the models could have preferred a much

different change time? In other words, since there has already been

claims of a change of IFR in the period considered, could this "a

posteriori information" be biasing the models?

4 - The full list of parameters for the models (at least A0) should

be more explicitly written down in an appendix.

6. PLOS authors have the option to publish the peer review history of their article (what does this mean?). If published, this will include your full peer review and any attached files.

Reviewer #1: No

Reviewer #2: No

---

## [Author Response · Author response to Decision Letter 0]

24 Aug 2021

Please see attached file reply.pdf

---

## [Decision Letter · Decision Letter 1]

11 Oct 2021

Bayesian inference across multiple models suggests a strong increase in lethality of COVID-19 in late 2020 in the UK

PONE-D-21-08131R1

Dear Dr. Pietzonka,

We’re pleased to inform you that your manuscript has been judged scientifically suitable for publication and will be formally accepted for publication once it meets all outstanding technical requirements.

Kind regards,

Jeffrey Shaman

Academic Editor

PLOS ONE

Additional Editor Comments (optional):

Reviewers' comments:

Reviewer's Responses to Questions

**Comments to the Author**

1. If the authors have adequately addressed your comments raised in a previous round of review and you feel that this manuscript is now acceptable for publication, you may indicate that here to bypass the “Comments to the Author” section, enter your conflict of interest statement in the “Confidential to Editor” section, and submit your "Accept" recommendation.

Reviewer #2: All comments have been addressed

2. Is the manuscript technically sound, and do the data support the conclusions?

Reviewer #2: Yes

3. Has the statistical analysis been performed appropriately and rigorously? 

Reviewer #2: Yes

4. Have the authors made all data underlying the findings in their manuscript fully available?

Reviewer #2: Yes

5. Is the manuscript presented in an intelligible fashion and written in standard English?

Reviewer #2: Yes

6. Review Comments to the Author

Reviewer #2: The authors carried out a large revision which addressed the majority of my criticisms. They also provided detailed explanations in their response letter. I thus consider that the revised manuscript can now be accepted for publication.

I leave final minor comments that the authors may consider implementing in the final, published version.

1 - Even though it is well cited, I'm not sure it makes sense to cite the very controversial Ioannidis study without taking the time to properly assess its very important limitations. I suggest the authors consider alternative studies such as Meyerowitz-Katz et al IJID 2020, Hauser et al PLOS Medicine 2020, Roques et al Biology 2020, Salje et al, Science 2020, O’Driscoll et al Nature 2021, Marra et al IJID 2021.

2 - I understand IFR is highly dependent on age, but I still consider it would enhance the manuscript if some values were included in the main text. One could either perform a weighted average over the 6 age bins using estimates of age demographics or simply quote one or two age bins. Of course, the caveats regarding the absolute estimation of the IFR with the used data would need to be highlighted in these numbers.

3 - Maybe the authors should consider including in the main text a sentence summarizing heir response as to why a narrow 2-week window is considered not to affect their results.

4 - In the new Section 5 the authors should consider briefly mentioning that the diagonal terms of the inverse Hessian already provides the credible intervals which include marginalization over all other parameters.

7. PLOS authors have the option to publish the peer review history of their article (what does this mean?). If published, this will include your full peer review and any attached files.

Reviewer #2: No

---

## [Editor Report · Acceptance letter]

28 Oct 2021

PONE-D-21-08131R1 

Bayesian inference across multiple models suggests a strong increase in lethality of COVID-19 in late 2020 in the UK 

Dear Dr. Pietzonka:

I'm pleased to inform you that your manuscript has been deemed suitable for publication in PLOS ONE. Congratulations! Your manuscript is now with our production department. 

Kind regards, 

on behalf of

Prof. Jeffrey Shaman 

Academic Editor

PLOS ONE